# Linking Pollution and Viral Risk: Detection of Dioxins and Coronaviruses in Cats and Dogs

**DOI:** 10.3390/v17091271

**Published:** 2025-09-19

**Authors:** Francesco Serra, Silvia Canzanella, Sergio Brandi, Gerardo Picazio, Anna Maria Pugliese, Luca Del Sorbo, Gianluca Miletti, Enza Ragosta, Emanuela Sannino, Filomena Fiorito, Mauro Esposito, Esterina De Carlo, Giovanna Fusco, Maria Grazia Amoroso

**Affiliations:** 1Experimental Zooprofilactic Institute of Southern Italy, 80055 Portici, Italy; silvia.canzanella@izsmportici.it (S.C.); sergio.brandi@izsmportici.it (S.B.); gerardo.picazio@izsmportici.it (G.P.); annamaria.pugliese@izsmportici.it (A.M.P.); gianluca.miletti@izsmportici.it (G.M.); enza.ragosta@izsmportici.it (E.R.); emanuela.sannino@izsmportici.it (E.S.); esterina.decarlo@izsmportici.it (E.D.C.); giovanna.fusco@izsmportici.it (G.F.); 2National Reference Center for the Analysis, and the Correlation Study Among Environment, Animals and Humans, Experimental Zooprofilactic Institute of Southern Italy, 80055 Portici, Italy; 3Department of Veterinary Medicine and Animal Production, University of Naples Federico II, 80137 Naples, Italy; lucadelsorbo00@gmail.com

**Keywords:** coronaviruses, dioxins, increased virulence, pollution, veterinary viruses

## Abstract

Viral and chemical analyses were performed on 80 dead cats and 51 dead dogs from the Campania Region (Southern Italy), with the aim of evaluating in vivo the potential correlation between coronavirus (CoV) infections and levels of environmental pollutants such as dioxins and PCSs (PCDD/F, DL-PCB and NDL-PCB). The overall viral prevalence was 16.3% in cats and 23.5% in dogs. Both feline coronavirus (FCoV) and canine coronavirus (CCoV) were identified, with variable detection rates in all the other organs investigated, supporting studies that provide evidence of systemic viral spread. The highest prevalence of coronaviruses (CoVs) was observed in Naples (19.2% for FCoV; 30.7% for CCoV) and Caserta (11.1% for FCoV; 50.0% for CCoV), areas that include municipalities with the highest Municipality Index of Environmental Pressure (MIEP) scores. Chemical analyses showed that DL-PCBs were present at more elevated concentrations in CoV-infected dogs and cats than in non-infected animals, whereas ∑NDL-PCB and ∑PCDD/F were detected in greater amounts in non-infected subjects. Among PCDDs, the congener 2,3,7,8-TCDD displayed different distribution patterns between infected and non-infected animals. In cats, 70.0% of FCoV-positive individuals had 2,3,7,8-TCDD levels above the limit of quantification (LOQ), compared with 38.0% of FCoV-negative cats. In dogs, 78.0% of CCoV-infected animals exceeded the LOQ, compared with 20.0% of non-infected ones; this difference was statistically significant. The results of the study suggest that elevated levels of 2,3,7,8-TCDD may be associated with CCoV infection and replication in dogs, suggesting a possible relationship between environmental pollution and susceptibility to coronavirus infections.

## 1. Introduction

Cats and dogs share the same living habitat as humans [1]. For this reason, they can provide valuable information about chemical and microbiological contamination of the environment and can help to find potential health risks for both humans and animals, serving as effective sentinels [2,3,4]. Living in close contact with humans poses a concrete risk of spillover events; this is particularly true for coronaviruses, which exhibit a high capacity for genetic variation, also gaining the ability to cross species barriers [5]. Polychlorinated dibenzo-p-dioxins (PCDDs) and polychlorinated dibenzofurans (PCDFs), a large family of aromatic organochlorine congeners, are persistent environmental contaminants due to their strong resistance to photo-degradation, metabolic and microbial processes. PCDD/PCDFs are mainly generated as by-products of industrial processes: they can be released into the environment during the production of pesticides, fungicides, herbicides, insecticides and wood preservatives, as well as during the chlorine bleaching of paper pulp. However, the most important source of PCDD/PCDF emissions is represented by municipal-, medical- and hazardous-waste-incineration plants [6,7,8,9,10,11,12,13,14,15,16,17,18]. Epidemiological studies on populations living near municipal-solid-waste and hazardous-waste incinerators have proven that exposure to their emissions may cause severe health effects, including cancer. Furthermore, PCDD/PCDFs are also released into the atmosphere through illegal incineration of industrial and urban waste. A striking example is the Italian area known as the “Terra dei Fuochi” (Land of Fire) in the Campania Region (Southern Italy), which had been the focus of both scientific investigation and legal proceedings [9,19,20,21,22,23]. Among PCDD/PCDFs, 17 congeners are of particular concern due to their high toxicity in both animals and humans. Due to this, in 2005 the World Health Organization established toxicity equivalency factors (TEFs) for these compounds [9]. Humans are primarily exposed to PCDD/PCDFs through the ingestion of food of animal origin, such as milk, dairy products, eggs, fish and meat [9,24]. Dietary intake accounts for approximately 90% of the total exposure, whereas inhalation contributes to the remaining 10% [24]. These compounds, being lipophilic, undergo biomagnification in the food chain, accumulate in adipose tissues and consequently cause toxicity in both humans and animals [6,7,8,9,17,18,25]. The congener 2,3,7,8-tetrachlorodibenzo-p-dioxin (2,3,7,8-TCDD) is the most toxic member of the PCDD/PCDF family. Indeed, based on evidence of cancer promotion in breast, thyroid and prostate tissues [26], it has been classified as a Group 1 human carcinogen by the International Agency for Research on Cancer (IARC). PCDD/PCDFs may induce endocrine disruption, liver toxicity, dermatological disorders, neuro-developmental impairment, reproductive organ toxicity and teratogenicity. Furthermore, they can cause thymic atrophy and generalized immunosuppression, leading to increased host susceptibility to viral infections [8,27,28,29,30]. Specifically, experimental studies have proven that TCDD exposure increases morbidity and/or mortality of mice infected with different influenza virus subtypes [31,32,33]. TCDD also enhances human immunodeficiency virus-1 (HIV-1) replication in various cell lines (MT-4 and U1) [34,35]. Similar findings have been reported with different herpesvirus, including cytomegalovirus (CMV) in MRC-5 cells [36] and bovine herpesvirus 1 (BoHV-1) in bovine MDBK cells [27,37,38,39]. Notably, dioxin exposure seems to affect coronavirus replication in vitro, as shown for murine hepatitis virus (MHV) in murine macrophages (BMDMs) [40], and for CCoV, specifically the CCoV type II strain S/378 (GenBank accession number KC175341), in A72 canine fibrosarcoma cells [41]. Canine coronavirus (CCoV), belonging to the family *Coronaviridae*, genus alphacoronavirus (α-CoV), is a positive-sense, single-stranded RNA virus infecting mammals and birds. Coronaviruses (CoVs), through ecological processes, possess the ability to cross interspecies barriers and undergo genetic adaptation. The accumulation of mutations in alternative hosts can promote the emergence of novel strains, sometimes characterized by alterations in both virulence and tissue tropism [42,43,44]. CCoV types I and II, as well as feline coronavirus (FCoV) types I and II, are common agents of enteric, respiratory and systemic infections [42,43,44,45,46,47]. Nevertheless, highly virulent CCoV-II strains were found to be responsible for fatal infections in puppies [48,49,50]. Recently, a new highly pathogenic recombinant FCoV-CCoV strain emerged, causing a feline infectious peritonitis outbreak in Cyprus [51,52]. Importantly, CoV canine–feline-like recombinant strains such as CCoV-HuPn-2018 and HuCCoV_Z19Haiti have recently been isolated from humans [53,54]. Our study reports the findings of the “CoV-Diox” project, funded by the Italian Ministry of Health within the framework of the “Land of Fire” issue. The project was undertaken to address health concerns arising from the prolonged illegal dumping and open-air combustion of toxic waste in the Campania Region. This territory had previously been analyzed to characterize the degree of pollution at the municipality level [55]. In that study the authors developed a mathematical model to compute, for each municipality, a synthetic index of environmental pressure, called MIEP, with values ranging from 0 (no environmental pressure) to 100 (maximum environmental pressure). The primary goal of the present research, a post-mortem tissue-based study, was the implementation of systematic surveillance of CoV infection in dogs and cats while also assessing the presence of chemical contaminants (PCDD/F, DL-PCB and NDL-PCB congeners) in the tissues of the same animals. The investigation was carried out on dead animals collected by local veterinarian authorities in the provinces of the Campania Region (including municipalities with different MIEP values) [55]. The final aim was to evaluate potential correlations between coronavirus infection and chemical contamination.

## 2. Materials and Methods

### 2.1. Sample Collection

The Annual Programming Document (DPAR) of the Campania Region—concerning official controls in the field of urban veterinary hygiene and animal management—defines and plans inspection, sampling, and surveillance activities to be carried out annually by territorial veterinary units. Within the DPAR, there is a mandatory activity whereby at least 5% of the dogs and cats found dead in the regional territory must undergo necropsy (carried out by the Zooprofilactic Experimental Institute of Southern Italy, IZSM) to ascertain the cause of death. Consequently, between 2021 and 2023 a total of 131 animal carcasses (51 dogs and 80 cats) were submitted to IZSM for necropsy. For each animal, background information, including age, sex, lifestyle, and geographical location, was recorded from the animals’ clinical history, although no data regarding clinical signs observed in life were available. Number of animals collected per province and area, including municipalities with the highest MIEP scores, is indicated in Figure 1. During necropsy, carried out by veterinarians and laboratory technicians, anatomopathological lesions compatible with infectious diseases were documented. Furthermore, within the CoV-Diox project, a small piece of each organ (lungs, liver, spleen, intestine) was sampled and investigated, via molecular assays, to evaluate the presence of CoV RNA. Moreover, in a subset of 32 liver samples (14 from dogs and 18 from cats), the concentration of 17 PCDD/F, 12 DL-PCB and 6 NDL-PCB congeners was determined.

### 2.2. Virological Analysis

#### 2.2.1. Viral Nucleic Acids Extraction

Tissue samples (25 mg) were individually homogenized by a Tissue Lyser (Qiagen GmbH, Hilden, Germany) at 30 Hz for 5 min in a 2 mL tube containing 1 mL of phosphate-buffered saline (PBS) and a 4.8 mm stainless steel bead. The homogenates (200 μL) underwent nucleic acid extraction with the QIAsymphony automated extraction system (Qiagen GmbH, Hilden, Germany) using a DSP Virus/Pathogen Mini kit (Qiagen GmbH, Hilden, Germany) according to the manufacturer’s instructions. Nucleic acids were eluted in 80 μL of elution buffer and immediately analyzed or stored at −80 °C until use. Quality assurance and control (QA/QC) were ensured by analyzing each sample in duplicate, as well as by including specific controls during both the extraction and amplification steps. Specifically, a negative process control, made with 200 μL PBS, was added at each extraction run. Furthermore, an external amplification control, murine norovirus, was spiked into each sample prior to extraction to evaluate possible PCR inhibition [56]. After extraction, samples underwent specific Real-Time RT-PCR assays for the detection of CCoV and FCoV. The two assays, routinary employed by our laboratory, underwent a validation process checked by the Quality System in compliance with UNI CEI EN ISO/IEC requirements and ISO 9001 guidelines.

#### 2.2.2. Real-Time PCR for the Detection of Feline Coronavirus

FCoV molecular detection was carried out by Real-Time RT-PCR using a QuantStudio 5 Real-Time PCR thermal cycler (Thermo Fisher Scientific, Waltham, MA, USA). The assay targeted a conserved 171-nucleotide region spanning the membrane–nucleocapsid gene junction of the strain FIPV 79-1146 [57]. Reactions were carried out in a 25 µL solution containing 5 µL nucleic acid extract, 12.5 µL AGPATH reaction buffer, 1 µL reverse transcriptase enzyme (Thermo Fisher Scientific, Waltham, MA, USA), 1 µL (6.25 µM) forward primer FCoV-For (5′-AGCAACTACTGCCACRGGAT-3′), 1 µL (6.25 µM) reverse primer FCoV-Rev (5′-GGAAGGTTCATCTCCCCAGT-3′) and 1 µL (5 µM) probe FCoV-P (5′-FAM-AATGGCCACACAGGGACAACGC-MGB-3′). The thermal profile was as follows: reverse transcription at 48 °C for 30 min and initial denaturation at 95 °C for 15 min, followed by 45 cycles of 95 °C for 15 s and 60 °C for 60 s [57]. A reference strain of FCoV type I, biotype FECV (isolate “Munchen”), kindly provided by FLI (Frederich Loeffer Institut, Greifswald-Insel Riems, Germany), was used as a PCR positive control (PC). Assay performance was monitored through quality control charts of the PC, updated with each analytical session. Sensitivity of the assay, calculated by standard curve analysis (built by amplifying serial dilution of the positive control) was 10^3^ TCID_50_/µL (Ct = 35). Samples with Ct values > 35 were considered negative.

#### 2.2.3. Real-Time PCR for the Detection of Canine Coronavirus

CCOV detection was performed via a Real-Time RT-PCR assay targeting a conserved 99 bp fragment of the ORF5 gene [58]. The reaction (total volume 25 µL), carried out using a QuantStudio 5 thermal cycler (Thermo Fisher Scientific, Waltham, MA, USA), contained 5 µL nucleic acid extract, 12.5 µL AGPATH reaction buffer, 1 µL reverse transcriptase enzyme (Thermo Fisher Scientific, Waltham, MA, USA), 1 µL (10 µM) forward primer CCoV-For (5′-TTGATCGTTTTTATAACGGTTCTACAA-3′), 1 µL (10 µM) reverse primer CCoV-Rev (5′-AATGGGCCATAATAGCCACATAAT-3′) and 1 µL (6 µM) probe CCoV-P (FAM-5′-ACCTCAATTTAGCTGGTTCGTGTATGGCATT-3′-TAMRA). The thermal profile was as follows: reverse transcription at 42 °C for 30 min and initial denaturation at 95 °C for 15 min, followed by 40 cycles of 95 °C for 15 s and 60 °C for 60 s [58]. CCoV type II strain CCoV-1-71, kindly provided by FLI (Frederich Loeffer Institut, Greifswald-Insel Riems, Germany), served as the PC. Assay performance was monitored through quality control charts of the PC, updated with each analytical session. Assay sensitivity, calculated as described above for FCoV, was 3.5 × 10^2^ TCID_50_/µL (Ct = 35). Samples exhibiting a Ct value > 35 were classified as negative.

#### 2.2.4. Molecular Typing of CCoV

CCoV-positive samples were further characterized by two separate Real-Time RT-PCR assays, specific to the CCoV-1 and CCoV-2 ORF5 genes. The reactions, carried out using a QuantStudio 5 thermal cycler (Thermo Fisher Scientific, Waltham, MA, USA) (final volume 25 μL), included 5 μL extracted RNA, 12.5 μL AgPath-IDTM One-step RT-PCR, 1 μL retro-transcriptase enzyme, 0.75 μL (10 μM) forward primer (CCoV1-For: 5′-CGTTAGTGCACTTGGAAGAAGCT-3′; CCoV2 For: 5′-TAGTGCATTAGGAAGAAGCT-3′) (Eurofins Genomic), 0.75 μL (10 μM) reverse primer (CCoV1-Rev: 5′-ACCAGCCATTTTAAATCCTTCA-3′; CCoV2 Rev: 5′-AGCAATTTTGAACCCTTC-3′) (Eurofins Genomic) and 0.83 μL (6 μM) probe (CCoV1-P: FAM-5′-CCTCTTGAAGGTACACCAA-3′-TAMRA; CCoV2-P: FAM-5′-CCTCCTGAAGGTGTGCC-3′-TAMRA) (Applied Biosystem by Thermo Fisher Scientific). The cycling conditions were 42 °C for 30 min and 95 °C for 15 min, followed by 45 cycles of 95 °C for 15 s, 53 °C (CCoV-I) or 48 °C (CCoV-II) for 30 s and 60 °C for 60 s [58].

#### 2.2.5. Digital Droplet PCR (ddPCR)

Samples testing positive in Real Time RT-PCR underwent ddPCR to quantify viral load. Droplet generation was performed with an Automated Droplet Generator (Bio-Rad, Hercules, CA, USA), according to the manufacturer’s instructions. PCR was run using a T100TM thermocycler, (Bio-Rad), employing the same primers and probe used in the Real-Time RT-PCR assays. The reactions (22 μL) included 5 μL RNA template, 5 μL supermix, 1 μL DTT, 2 μL reverse transcriptase (One-Step RT-ddPCR; Bio-Rad), 4 μL DNAse/RNAse-free water, 2 μL of each primer (final concentration of 900 nM) and 1 μL of probe (final concentration of 250 nM). The thermal profile was as follows: 50 °C for 60 min and 95 °C for 10 min, followed by 45 cycles of 95 °C for 15 s and 60 °C for 60 s, and final enzyme inactivation at 98 °C for 10 min. The results were analyzed using a QX200 Droplet Reader (Bio-Rad).

#### 2.2.6. Virus Isolation

Samples with Ct ≤ 30 were subjected to virus isolation. Specifically, 0.5 g of tissue sample was homogenized by mortar and pestle with 0.1 mm diameter silica sand (Benchmark Silica Glass Bulk Beads, Sigma-Aldrich, St. Louis, MO, USA) in 5 mL EMEM (Gibco, Life Technologies, Europe B.V., Bleiswijk, The Netherlands). Homogenates were clarified by centrifugation (1500× *g* for 10 min) and filtered (0.45 nm filters, Millipore, Thermo Fisher Scientific, Waltham, MA, USA). A-72 (ATCC CRL-1542) and CRFK (ATCC CCL-94) cells were, respectively, used for CCoV and FCoV infection. Cells were cultured in EMEM (Gibco, Life Technologies, Europe B.V., Bleiswijk, The Netherlands) supplemented with 10% FBS (Gibco), 1% antibiotic–antimycotic (Gibco) and 1% l-glutamine (Gibco) at 37 °C with 5% CO_2_. Confluent cell monolayers were inoculated with 100 µL homogenate, incubated for 1 h and then replenished with maintenance medium (0.2% FBS, 1% antibiotic–antimycotic, 1% l-glutamine). Control cultures were prepared by using a pure non-supplemented EMEM inoculum. Cultures were monitored daily by inverted microscopy for up to 7 days to ascertain whether there was a cytotoxicity and cytopathic effect (CPE). Every week, for a total of 3 weeks, cell subcultures were carried out using 100 µL supernatant as the inoculum. When CPE was observed, viral RNA was extracted from cell supernatants and confirmed by a specific Real-Time RT-PCR assay [59].

### 2.3. Chemical Analysis

#### 2.3.1. PCDD/F and DL-PCB Analysis

Liver samples were analyzed to detect and quantify 7 PCDDs, 10 PCDFs and 12 DL-PCBs as specified by Commission Regulation (EU) No. 2017/644. The procedure followed adaptations of US EPA Methods 1613 Revision B (1994) and 1668 Revision C (2010) (see website: www.epa.gov). Samples were first thawed to room temperature and homogenized. Aliquots of 5–15 g (depending on availability) were taken to ensure sufficient material for fat extraction, with a target yield of 1–5 g of fat. The homogenized samples were frozen at −20 °C for 24 h, lyophilized and spiked with an internal standard solution containing ^13^C_12_-isotope-labeled PCDD/Fs and PCBs. Lipid extraction was conducted using an accelerated solvent extraction system (Dionex ASE 350, Thermo Fisher Scientific) with a hexane/acetone mixture (4:1, *v*/*v*). The extract underwent initial purification using an acidified multilayer column, followed by a three-step clean-up process with an automated Power Prep^®^ system (FMS—Fluid Management Systems, Inc., Billerica, MA, USA) employing silica, alumina and carbon columns. The fraction containing mono-ortho DL-PCBs was eluted from the alumina column, while the fraction containing PCDD/Fs and non-ortho PCBs was eluted from the carbon column. Both fractions were concentrated under nitrogen and then under vacuum. The residues were subsequently dissolved in recovery standard solutions (^13^C_12_-labeled congeners) and analyzed by a high-resolution gas chromatograph coupled to a high-resolution mass spectrometer (DFS Magnetic Sector HRGC-HRMS system, Thermo Fisher Scientific).

#### 2.3.2. NDL-PCB Analysis

For NDL-PCB determination, 0.5 g liver tissue was spiked with a standard solution containing six congeners of ^13^C_12_-labeled NDL-PCBs (IUPAC 28, 52, 101, 138, 153 and 180; Cambridge Isotope Laboratories, Tewksbury, MA, USA) to evaluate recovery. Lipid content was extracted with diethyl ether (Carlo Erba Reagents, Milan, Italy) over 24 h. Extracts were filtered through Whatman filter paper with anhydrous sodium sulfate and dried using a rotary evaporator. Clean-up was performed on columns packed with acidified diatomaceous earth (Merck KGaA, Darmstadt, Germany) and Florisil SPE cartridges (Biotage, Uppsala, Sweden). The eluate was concentrated to approximately 0.2 mL, transferred to gas chromatography vials with glass inserts and spiked with 2 µL tetradecane. After solvent evaporation under vacuum, the residue was reconstituted in 50 µL injection standard solution (^13^C_12_-labeled congeners) for NDL-PCB determination and analyzed by HRGC-HRMS.

#### 2.3.3. Instrumental Analysis

Identification and quantification of PCDD/Fs, DL-PCBs and NDL-PCBs were performed using a high-resolution gas chromatograph paired with a high-resolution mass spectrometer (HRGC-HRMS, DFS Magnetic Sector GC-HRMS system, Thermo Fisher Scientific), using the isotope dilution method. Samples were injected in splitless mode, with an injector and transfer line temperature of 280 °C. Chromatographic separation of PCDDs, PCDFs and four non-ortho DL-PCB congeners (77, 81, 126 and 169) was achieved using a TR-5MS fused-silica capillary column (60 m length, 0.25 mm inner diameter, 0.1 µm film thickness, Thermo Fisher Scientific) coated with 5% phenyl polysilphenylene-siloxane. Separation of eight mono-ortho PCB congeners (105, 114, 118, 123, 156, 157, 167 and 189) and six NDL-PCB congeners (28, 52, 101, 153, 138 and 180) was performed using an HT 8 fused-silica capillary column (60 m length, 0.25 mm inner diameter, 0.25 µm film thickness, SGE Analytical Science) coated with 8% phenyl polysiloxane. In both chromatographic methods, helium (99.9999% purity), at a constant flow of 1.0 mL/min, was used as a carrier gas. Operating at a resolution of 10,000 (10% valley), the mass spectrometer collected data by monitoring two isotopic masses for each PCDD/F and PCB congener. Congener identification was performed by comparing the analytes’ chromatographic retention times with ^13^C-labeled internal standards and confirming that the selected ion pairs for each congener group met the relative abundance ratios established in the calibration standards. QA/QC was ensured by including blanks, spikes and duplicates, and by participating in interlaboratory studies, with z-scores within the range of ±2. Recovery performance was monitored through quality control charts of ^13^C 2,3,7,8-TCDD, ^13^C 2,3,4,7,8-PeCDF and ^13^C PCB-126 updated with each analytical session.

### 2.4. Statistical Analysis

Statistical analysis was performed using GraphPad Software InStat 3 (Dotmatics, Atlanta, GA, USA). Percentages with 95% confidence intervals (CI) were calculated using the modified Wald method. Comparison of categorical variables was carried out with Fisher’s exact test. For chemical data, most variables did not follow a standard distribution (Shapiro–Wilk test; *p*-value < 0.05). Therefore, non-parametric tests were used. The Mann–Whitney U test was employed to assess differences in PCDD/F (including 2,3,7,8-TCDD), DL-PCB and NDL-PCB concentrations between animals infected and not infected by coronavirus. Values below the limit of quantification (LOQ) were set to half the LOQ. The non-parametric Spearman test was employed to evaluate correlations between individual congeners of PCDD/F, DL-PCB and NDL-PCB, their sums, and CoV positivity. A *p*-value of ≤0.05 was considered significant in all tests.

## 3. Results

### 3.1. Positivity for Feline Coronavirus (FCoV)

Real Time RT-PCR for the detection of FCoV, carried out on 80 cats, showed 16.3% (13/80) positivity with threshold cycle (Ct) values ranging from 22 to 34. The virus was detected in all types of tissues analyzed, with the following prevalence: intestine 11/13 (84.6%), lungs 7/13 (53.8%), liver 7/13 (53.8%) and spleen 4/13 (30.8%) (Figure 2). In 7/13 cats (53.8%) the virus was uncovered in more than one organ. As indicated in Table 1, all 13 FCoV-infected cats exhibited, during necropsy, lesions consistent with infectious disease. Specifically, 11 animals showed anatomopathological lesions compatible with enteritis, 4 of which also presented lesions attributable to bronchopnuemonia. In two cats the lesions were instead exclusively associated with pneumonia. As expected (Table 1), FCoV was detected in the intestines of 10/11 animals with enteric lesions. Interestingly, one animal with enteritis harbored the virus only in the kidney but not in the gastrointestinal tract. Among the six cats exhibiting respiratory tract lesions, FCoV was revealed in the lungs of four. Furthermore, in 4/13 animals (30.8%) the virus appeared to be systematically distributed, being detected in all organs examined. All the positive samples, quantified by ddPCR, showed a viral load ranging from 0.26 copies/μL to 1905 copies/μL. The highest concentration was observed in the intestine. The mean viral loads per organ were as follows: intestine and liver, 358 copies/μL; lungs, 326 copies/μL; and spleen, 15 copies/μL. The data were further analyzed according to sex, age (kitten vs. junior/adult), breed, lifestyle (stray vs. owned) and province of collection (Naples, Avellino, Benevento, Caserta, Salerno) (Table 2). None of these variables showed statistically significant associations with FCoV infection, probably due to the different sizes of the groups. However, when comparing homogeneous groups by sex (Table 2), a higher prevalence was observed in females than in males (22.5% vs. 10%, *p* = 0.22, see Table 2).

### 3.2. Positivity for Canine Coronavirus (CCoV)

Real Time RT-PCR revealed the presence of CCoV in 12 out of 51 dogs analyzed (23.5%)*,* with Ct values from 23 to 35. All the identified instances of CCoV were further classified as type II canine coronavirus. CCoV was detected in all the examined organs, with higher prevalence in the intestine (66.7%, 8/12) and lungs (58.3%, 7/12) compared to the liver (41.7%, 5/12) and spleen (33.3%, 4/12) (Figure 2). In 7/12 dogs (58.3%) the virus was present in more than one organ, with a systemic distribution (virus detected in all organs) observed in 2 animals. Necropsy revealed that all the CCoV-infected dogs displayed lesions compatible with coronavirus infection (gastroenteric and/or respiratory lesions). However, two animals since found to be poisoned by dicoumarins or metaldehyde were excluded from further investigations. Of the other 10 CCoV-infected animals, 6 showed anatomopathological lesions compatible with enteritis, 4 of which also exhibited lesions attributable to bronchopnuemonia. In four animals lesions were exclusively associated with pneumonia (Table 3). As expected, CCoV was found in the intestines of nearly all animals with enteric lesions (5/6). Interestingly, in one animal with enteritis the virus was uncovered in the lungs but not in the gastrointestinal tract. In six of the seven dogs with respiratory lesions, CCoV was identified in the lungs (Table 3). Viral load, measured by ddPCR, varied from 0.2 copies/μL to 77.6 copies/μL, with the highest concentrations found in the intestine. The average viral loads per organ were as follows: intestine, 31 copies/μL; liver, 24 copies/μL; spleen, 4 copies/μL; and lungs, 3 copies/μL. The data were further analyzed according to sex, age, breed, lifestyle and province of collection (Table 4).

The results showed a statistically significant difference in CCoV infection between junior/adult and young dogs (12.8 vs. 58.3%, *p* < 0.01, O.R. 10). For the other variables investigated (sex, breed, lifestyle and province) interesting data (even if not statistically significant) were obtained. Specifically, females appeared to be more susceptible to coronavirus compared to males (33.3% vs. 16.6, *p* = 0.19). Furthermore, a higher incidence of CCoV was observed in purebred dogs with respect to mixed-breed dogs (34.7% vs. 14.2%, *p* = 0.10), and strays dogs appeared to be more frequently infected than owned dogs (34.6% vs. 12.0%, *p* = 0.09).

### 3.3. Viral Isolation by Cell Culture

All samples (both canine and feline) with Ct < 30 in Real-Time RT-PCR underwent viral isolation, with the aim of further studying (through in vitro experiments) and characterizing (through sequencing) the strains detected. The results were unfortunately inconclusive. Indeed, when a cytopathic effect was observed, the cell supernatant tested negative in specific Real-Time RT-PCR. Lack of isolation did not change the interpretation of the results: samples that tested positive in Real-Time RT-PCR were considered positive for both the statistical analysis and chemical investigations.

### 3.4. Presence of Contaminants in Cats and Dogs Infected/Not Infected by Coronaviruses

Chemical analyses for the determination of 17 PCDD/F congeners (including 2,3,7,8-TCDD) and 12 DL-PCB congeners were performed on 32 liver samples, 14 from dogs and 18 from cats. Of the 14 dog livers, 9 were infected with CCoV, while 5 were CoV-negative.

Among the 18 cat livers, 10 were infected with FCoV, while 8 were CoV-negative. For PCDD/F and DL-PCB, the congener concentrations were obtained, and are reported in Table 5 as the mean ± standard deviation, in pg/g of wet weight (w.w.). The sum of PCDD/F and DL-PCB in the samples is expressed as pg-TEQ/g (toxic equivalent). For NDL-PCB, individual congener concentrations (see Table 6) were summed and are expressed in ng/g of wet weight (w.w.). Total concentrations of PCDD/F, DL-PCB and NDL-PCB were calculated using the upper bound approach. This method assumes that the contribution of each non-detectable congener to the TEQ summation is equal to its respective limit of quantification.

The results related to the sums of DL-PCB, NDL-PCB and PCDD/F (Table 5 and Table 6) yielded contrasting outcomes with respect to CoV infection. In detail, ∑NDL-PCB and ∑PCDD/F were detected in higher amounts in non-infected animals (both cats and dogs) compared to infected animals, whereas DL-PCB was detected at higher levels in CoV-infected dogs and cats compared to non-infected animals. Interestingly, among the PCDDs, the concentrations of the congener 2,3,7,8 TCDD (reported in Table 7) showed different patterns in infected and non-infected animals. In cats, 70.0% of FCoV-infected liver samples had 2,3,7,8-TCDD levels above the limit of quantification (LOQ), compared to 38.0% of non-FCoV-infected cats. In dogs, 78.0% of CCoV-infected liver samples exceeded the LOQ, compared to 20.0% in non-CCoV-infected dogs. Statistical analysis using the Mann–Whitney U test revealed no significant differences in the concentrations of 2,3,7,8-TCDD between FCoV-infected and non-FCoV-infected cats (*p* = 0.98). In dogs, however, a statistically significant difference was observed, with higher concentrations of 2,3,7,8-TCDD in infected animals (*p* = 0.031). Spearman’s rank correlation test for cats did not show any correlation between the levels of PCDD/Fs, DL-PCBs and NDL-PCBs, their sums, and FCoV positivity. The only exception was the congener 1,2,3,4,7,8,9-HpCDF, whose levels showed a negative correlation with FCoV positivity (Spearman’s rs = −0.47; *p* = 0.047). In dogs, the Spearman correlation test indicated a moderate and significant correlation between 2,3,7,8-TCDD concentration and CCoV positivity (Spearman’s rs = 0.59; *p* = 0.031).

## 4. Discussion

Our study focused on coronavirus infection in dogs and cats living in areas characterized by different MIEP scores [55], yielding compelling results. FCoV prevalence showed a non-negligible positivity rate of 16.3%, consistent with previous findings in the same region (21.5%) [60], though lower than those documented in other studies conducted in Malaysia (84.0%) [61], Italy (29.6%) [62] and China (67.9%) [63]. Such discrepancies are not surprising and may be due to several factors, including differences in sample size and geographical distribution. When evaluating risk factors that might influence FCoV prevalence, none of the variables investigated (sex, age, lifestyle or province) showed statistically significant associations, probably due to uneven group sizes. Sex was the only variable for which the two groups were homogeneous; however, the higher prevalence observed in females compared to males (9/40 vs. 4/40), although not statistically significant, is inconsistently corroborated in the literature [60,61,62,63,64], suggesting that sex cannot be considered a predictive factor for FCoV infection. Regarding viral distribution, our results confirmed, as widely reported in the literature, that the intestine is the primary site of FCoV replication and persistence [65,66,67]. The virus was, however, identified with variable percentages in all the other organs examined, supporting studies that provide evidence of systemic viral spread [67,68]. Interestingly it was found in the lungs of four out of six animals showing lesions compatible with bronchopneumonia, reinforcing the hypothesis that FCoV may also be implicated in respiratory symptomatology [67,69]. Further studies, including in vitro experiments with the field strains identified, would be necessary to confirm this association. Unfortunately, our viral isolation experiments (for both coronaviruses under study) gave inconclusive results, probably due to abortive replication, which has already been documented in the literature for these viruses [70,71,72,73,74]. In dogs, CCoV prevalence was 23.5%, consistent with the global range of 17.4–25.6% [75] and close to the 24.4% reported for European countries [75]. In Italy, there are only a few studies documenting CCoV prevalence. Among them, Mira et al. [76] reported a positivity rate of 13.7% in dogs from the Sicily Region, while Cardillo et al. [77] found a positivity rate of 31.1% in the Campania Region. Compared to the latter, we observed a lower percentage of infected animals in the same geographical area, probably because the authors analyzed only young dogs that are more susceptible to the virus than adults. In fact, the positivity rate we calculated was significantly higher in dogs of 0–6 months (58.3%) compared to adults (13.5%), confirming that CCoV primarily infects puppies [78,79]. CCoV is a common agent of gastroenteritis in dogs [76,79,80], typically associated with high morbidity but low mortality [81]. Accordingly, in 6 of the 10 infected animals, we found anatomopathological lesions compatible with enteritis. However, CCoV has also been linked to severe clinical signs and fatal events [79,82,83]. This can occur in cases of co-infection with other pathogens [81,84], as well as in single infections caused by highly virulent strains [85,86,87,88]. In this regard, all instances of CCoV identified by Real-Time RT-PCR in our study were characterized as type II canine coronavirus. Consistent with the literature [48,70,76,89], CCoV-II was not only detected in the gastrointestinal tract, but also in organs like the kidney, spleen, and lungs, corroborating its broader tissue tropism. Remarkably, in 8 of the 10 infected dogs, CCoV-II was identified in the lungs, and all these animals showed bronchopneumonia-like lesions, suggesting the circulation of hypervirulent strains. Even though co-infections cannot be excluded (not ascertained since this was not the focus of this research), there is substantial scientific literature reporting the ability of CCoV-II to cause systemic infections [79,82,83]. It is noteworthy that increased pathogen virulence may be related to animal exposure to chemical contaminants. In this regard, it has been shown that 2,3,7,8-TCDD can induce thymic atrophy and immune suppression, thereby increasing susceptibility to viral infections [8,27,28,29,30]. Furthermore, in mice exposed to 2,3,7,8-TCDD, infection with mouse hepatitis virus (MHV) [90] or with various influenza virus subtypes, resulted in increased morbidity and mortality [31,32,33]. Moreover, 2,3,7,8-TCDD enhanced the in vitro replication of MHV [40], human immunodeficiency virus-1 [34,35], cytomegalovirus [36] and bovine herpesvirus 1 [6,37,38,39]. Our results showed, in dogs, a moderate but significant correlation between chemical exposure and increased susceptibility to CCoV infection (Spearman’s rs = 0.59; *p* = 0.031). These findings are in accord with recent in vitro studies indicating that canine coronavirus infection is intensified by 2,3,7,8-tetrachlorodibenzo-p-dioxin [41]. Furthermore, our results showed the highest prevalence of positivity for CoVs in Naples (19.2% for FCoV; 30.7% for CCoV) and Caserta (11.1% for FCoV; 50.0% for CCoV). These provinces, known for illegal waste incineration activities, include municipalities with the highest MIEP scores [55], as also confirmed by the relative abundance of the congeners detected (see Table 5 and Table 6). These findings reinforce the hypothesis of a possible environmental link [21], suggesting that exposure to chemical contaminants may be somehow associated with companion animals’ susceptibility to CCoV and FCoV infections. This concept is supported by studies in humans showing that exposure to air pollutants such as PM2.5, PM10, SO_2_, NO_2_, O_3_ or CO is associated with increased incidence and severity of COVID-19 [91,92,93,94]. Moreover, several studies conducted in both European and non-European countries have reported correlations between environmental pollution and mortality rates from SARS-CoV-2 infection [91,92,94,95,96,97,98,99]. Although this study revealed only a weak correlation between dioxin levels and coronavirus infections, further investigations exploring the role of aryl hydrocarbon receptor (AhR) signaling may help clarify the mechanisms underlying this relationship. AhR is a transcription factor regulated by various substances, including environmental contaminants and microbial metabolites, and is responsive both to dioxins and PCBs, as well as to CoV infections [100,101]. AhR binds different endogenous and exogenous substances, forming a complex that, by modulating Cytochrome P450 enzymes levels, ultimately affects cytokine levels and immune function [100,101]. Notably, AhR has been shown to be up-regulated during infection with several coronaviruses, including Middle East respiratory syndrome coronavirus, human coronavirus (HCoV) 229E, HCoV-OC43, SARS-CoV-1, SARS-CoV-2, MHV and CCoV-II [40,90,102,103,104,105]. Interestingly, the selective AHR inhibitor CH223191 blocks MHV infection by lowering the expression of interleukin (IL)-1β and (IL)-10 while increasing tumor necrosis factor alpha levels, whereas a typical AhR activator such as TCDD reverses these effects [40].

## 5. Conclusions

Despite certain limitations of this study, including sample size, geographic scope, unsuccessful viral sequencing, and potential confounding variables, our findings suggest that high levels of 2,3,7,8-TCDD may be involved in CCoV replication in dogs, suggesting a possible link between environmental pollution and coronavirus infections. Since cats and dogs share their environment with humans, they are considered effective sentinels for the early detection of environmental hazards. From this point of view, further studies investigating a broader spectrum of environmental contaminants beyond dioxins and PCBs, and their impact on viral infectious diseases, could provide valuable insights for both animal and human health.

## Figures and Tables

**Figure 1 viruses-17-01271-f001:**
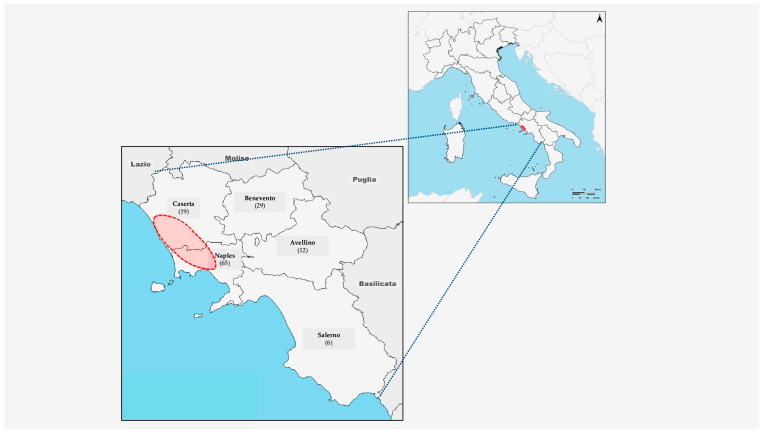
Provinces of the Campania Region (Southern Italy) and number of animals (dogs and cats) collected. Red circle indicates the area in which the municipalities with the highest MIEP scores (from 46 to 100) are concentrated (provinces of Naples and Caserta) [55].

**Figure 2 viruses-17-01271-f002:**
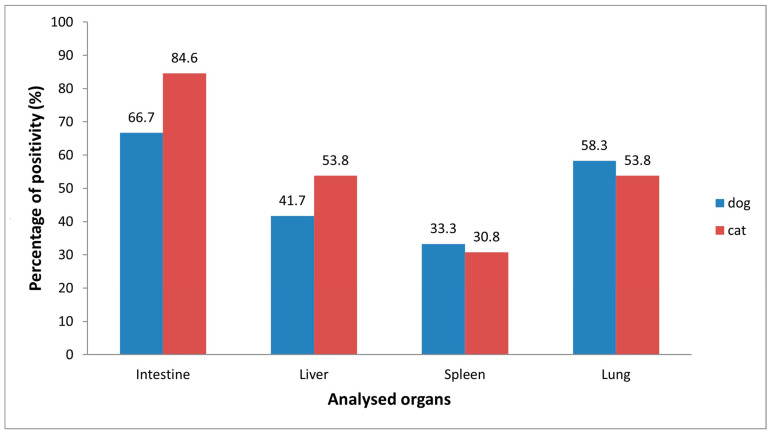
Coronavirus prevalence (%) in organs of animals (dogs and cats) found to be infected.

**Table 1 viruses-17-01271-t001:** Anatomopathological lesions detected in infected cats, and FCoV organ distribution. Necropsy signs were collected only post-mortem (no clinical history available). FCoV-infected cats: animals in which the virus was detected in at least one organ.

FCoV-Infected Cats (*n* = 13)	Anatomopathological Lesions Observed	Organs Positive for FCoV in Real Time RT-PCR
(1)	Enteritis and Pneumonia	Intestine, lungs, spleen, liver
(2)		Intestine, lungs, spleen, liver
(3)		Intestine, lungs, liver
(4)		Intestine
(5)	Pneumonia	Intestine
(6)		Lungs
(7)	Enteritis	Intestine, lungs, spleen, liver
(8)		Intestine, lungs, spleen, liver
(9)		Intestine, liver
(10)		Intestine, lungs
(11)		Intestine
(12)		Intestine
(13)		Liver

**Table 2 viruses-17-01271-t002:** FCoV prevalence in cats with respect to collection year, collection site (province) sex, age, breed and lifestyle.

Subgroups		*n*	Number of Infected Animals	*p* Value
**Sex**	Female	40	9 (22.5%) CI 12.1–37.7	*p* = 0.22
	Male	40	4 (10.0%) CI 3.3–23.6	
**Age ^1^**	Junior/Adult	62	11 (17.7%) CI 10.3–30.0	*p* = 0.72
	Kitten	18	2 (11.1%) CI 1.8–34.0	
**Breed**	Mixed-Breed	2	0 (0.0%) CI 0.0–70.9	*p* = 1.00
	Purebred *	78	13 (16.6%) CI 9.8–26.6	
**Lifestyle**	Stray	71	11 (15.4%) CI 8.7–25.8	*p* = 0.63
	Owned	9	2 (22.2%) CI 5.3–55.7	
**Province**	Naples	52	10 (19.2%) CI 10.6–32.0	*p* = 0.76
	Benevento	17	2 (11.7%) CI 2.0–35.5	
	Caserta	9	1 (11.1%) CI 0.0–45.6	
	Salerno	2	0 (0.0%) CI 0.0–70.9	
**Total**		80	13 (16.2%) CI 9.6–25.9	

Abbreviation: CI = confidence interval (95%). * Purebred cats were all European. Only 1 was Canadian (Sphynx). ^1^ Kitten = animals of 0–12 months; Junior/Adult = animals > 12 months.

**Table 3 viruses-17-01271-t003:** Anatomopathological lesions detected in infected dogs, and CCoV organ distribution. Necropsy signs were collected only post-mortem (no clinical history available).

CCoV-Infected Dogs (*n* = 12)	Anatomopathological Lesions Observed	Organs Positive for CCoV In Real Time RT-PCR
**(1)**	**Enteritis and Pneumonia**	Intestine, lungs, spleen, liver
**(2)**		Intestine, lungs, spleen, liver
**(3)**		Intestine, liver
**(4)**		Lungs
**(5)**	**Pneumonia**	Intestine, lungs, liver
**(6)**		Intestine, lungs
**(7)**		Intestine, lungs
**(8)**		Lungs
**(9)**		Spleen
**(10)**	**Enteritis**	Intestine, liver
**(11)**		Intestine
**(12)**		Spleen

CCoV-infected dogs: animals in which the virus was detected in at least one organ.

**Table 4 viruses-17-01271-t004:** CCoV prevalence in dogs with respect to collection year, collection site (province) sex, age, breed and lifestyle.

Animal Characteristics		*n*	Number of Infected Animals	*p* Value
**Sex**	Female	21	7 (33.3%) CI 17.0–54.7	*p* = 0.19
	Male	30	5 (16.6%) CI 6.8–34.0	
**Age ^1^**	Junior/Adult	39	5 (12.8%) CI 5.4–28.4	*p* < 0.01
	Young	12	7 (58.3%) CI 3.8–80.7	
**Breed**	Mixed-Breed	28	4 (14.2%) CI 5.0–32.1	*p* = 0.10
	Purebred *	23	8 (34.7%) CI 18.7–55.2	
**Lifestyle**	Stray	26	9 (34.6%) CI 19.3–53.8	*p* = 0.09
	Owned	25	3 (12.0%) CI 3.3–30.7	
**Province**	Naples	13	4 (30.7%) CI 12.3–57.9	*p* = 0.12
	Benevento	12	1 (8.3%) CI 0.0–37.5	
	Caserta	10	5 (50.0%) CI 23.6–76.3	
	Salerno	4	1 (25.0%) CI 3.4–71.0	
	Avellino	12	1 (8.3%) CI 0.0–37.5	
**Total**		51	12 (23.5%) CI 13.8–36.9	

Abbreviation: CI = confidence interval (95%). * Of the purebred dogs, 21 were European and 2 were Mexican (Chihuahua). ^1^ Young = animals of 0–6 months; junior/adult = animals > 6 months.

**Table 5 viruses-17-01271-t005:** Mean concentration and standard deviation of PCDD, PCDF and DL-PCB congeners (expressed in pg/g w.w.) and their sums (expressed in pg-TEQ/g), excluding outliers, in liver samples from dogs and cats, both positive (P) and negative (CN) for CoVs.

Liver	Dog (P)	Dog (CN)	Cat (P)	Cat (CN)
Congener	*n* = 9 Mean ± SD	*n* = 5 Mean ± SD	*n* = 10 Mean ± SD	*n* = 8 Mean ± SD
**PCDD**				
**2,3,7,8-TCDD**	**0.0066 ± 0.0073**	**ND**	**0.0088 ± 0.0093**	**0.0058 ± 0.0029**
12378-PeCDD	ND	ND	ND	0.038 ± 0.019
123478-HxCDD	0.061 ± 0.027	0.065 ± 0.028	0.055 ± 0.017	0.064 ± 0.026
123678-HxCDD	0.021 ± 0.0092	0.042 ± 0.029	0.016 ± 0.011	0.025 ± 0.020
123789-HxCDD	ND	ND	0.017 ± 0.019	0.014 ± 0.0075
1234678-HpCDD	0.44 ± 0.31	0.95 ± 1.6	0.25 ± 0.22	0.42 ± 0.61
OCDD	3.2 ± 4.3	2.5 ± 3.9	1.6 ± 1.4	1.7 ± 1.9
**PCDF**				
2378-TCDF	0.015 ± 0.0056	0.0068 ± 0.0041	0.020 ± 0.0086	0.024 ± 0.0086
12378-PeCDF	ND	ND	ND	0.016 ± 0.0095
23478-PeCDF	0.088 ± 0.057	0.19 ± 0.14	0.095 ± 0.052	0.12 ± 0.071
123478-HxCDF	0.083 ± 0.058	0.19 ± 0.15	0.051 ± 0.052	0.062 ± 0.027
123678-HxCDF	0.050 ± 0.034	0.13 ± 0.071	0.042 ± 0.046	0.049 ± 0.024
123789-HxCDF	0.0078 ± 0.0035	ND	0.024 ± 0.039	0.014 ± 0.0073
234678-HxCDF	0.036 ± 0.024	0.053 ± 0.026	0.084 ± 0.096	0.066 ± 0.051
1234678-HpCDF	0.12 ± 0.053	0.079 ± 0.055	0.20 ± 0.22	0.20 ± 0.070
1234789-HpCDF	0.016 ± 0.0060	0.021 ± 0.020	0.029 ± 0.050	0.022 ± 0.0098
OCDF	0.13 ± 0.074	0.080 ± 0.034	1.5 ± 4.0	0.15 ± 0.055
**DL-PCB**	*n* = 8	*n* = 4	*n* = 9	*n* = 6
PCB 77	1.3 ± 0.30	1.2 ± 0.51	1.5 ± 0.56	2.4 ± 1.7
PCB 81	0.13 ± 0.088	0.077 ± 0.041	0.12 ± 0.039	0.19 ± 0.12
PCB 126	0.73 ± 1.4	0.083 ± 0.066	0.25 ± 0.31	0.16 ± 0.074
PCB 169	0.14 ± 0.15	0.13 ± 0.064	0.19 ± 0.24	0.078 ± 0.023
PCB 105	7.4 ± 5.2	4.6 ± 1.1	50.2 ± 47.1	24.6 ± 13.1
PCB 114	0.84 ± 064	0.61 ± 0.23	5.9 ± 4.6	2.9 ± 0.99
PCB 118	23.2 ± 14.7	13.5 ± 3.8	142.5 ± 121.3	68.1 ± 27.6
PCB 123	0.50 ± 0.42	0.25 ± 0.084	0.79 ± 0.78	0.55 ± 0.38
PCB 156	4.1 ± 3.5	2.7 ± 0.33	45.8 ± 39.5	28.4 ± 33.9
PCB 157	1.5 ± 1.7	0.80 ± 0.88	10.7 ± 9.2	4.5 ± 2.8
PCB 167	2.6 ± 3.3	0.39 ± 0.12	11.6 ± 10.8	4.2 ± 2.5
PCB 189	1.4 ± 2.5	1.2 ± 0.55	7.9 ± 7.7	5.8 ± 9.5
**∑_17_PCDD/F**	0.070 ± 0.032	0.12 ± 0.081	0.076 ± 0.046	0.089 ± 0.057
**∑_12_DL-PCB**	0.078 ± 0.14	0.012 ± 0.0090	0.039 ± 0.038	0.022 ± 0.010
**∑PCDD/F + DL-PCB**	0.15 ± 0.17	0.15 ± 0.077	0.12 ± 0.081	0.11 ± 0.048

*n*: number of samples; (P): positive for CoVs; (CN): negative control; ND: not detectable; SD: standard deviation.

**Table 6 viruses-17-01271-t006:** Mean concentration and standard deviation of NDL-PCB congeners (expressed in ng/g w.w.) and their sums (expressed in ng/g w.w.), excluding outliers, in liver samples from dogs and cats, both positive (P) and negative (CN) for CoVs.

Liver	Dog (P)	Dog (CN)	Cat (P)	Cat (CN)
Congener	*n* = 8 Mean ± SD	*n* = 5 Mean ± SD	*n* = 8 Mean ± SD	*n* = 6 Mean ± SD
**NDL-PCB**				
PCB 101	0.62 ± 0.60	0.58 ± 0.54	0.23 ± 0.26	0.42 ± 0.46
PCB 28	0.90 ± 1.1	0.61 ± 0.82	0.34 ± 0.51	0.81 ± 0.77
PCB 153	0.13 ± 0.12	0.68 ± 1.3	0.45 ± 0.47	0.37 ± 0.27
PCB 52	1.1 ± 1.3	0.92 ± 1.4	0.37 ± 0.61	0.84 ± 1.0
PCB 138	0.083 ± 0.086	0.50 ± 0.93	0.31 ± 0.36	0.29 ± 0.30
PCB 180	0.056 ± 0.046	0.32 ± 0.60	0.22 ± 0.22	0.12 ± 0.071
**∑_6_NDL-PCB**	2.9 ± 3.2	3.6 ± 3.7	1.9 ± 1.9	2.9 ± 2.6

**Table 7 viruses-17-01271-t007:** Concentrations of the congener 2,3,7,8-TCDD (expressed in pg/g) in liver samples from cats and dogs that tested positive (P) and negative (CN) for CoVs.

Dog (P)	Dog (CN)	Cat (P)	Cat (CN)
*n* = 9	*n* = 5	*n* = 10	*n* = 8
0.0060	ND < 0.0004	0.029	ND < 0.0008
0.0070	0.0033	0.0025	0.0090
ND < 0.0007	ND < 0.0002	ND < 0.0003	ND < 0.002
0.0041	ND < 0.0003	0.0094	0.0033
0.00073	ND < 0.001	0.0055	ND < 0.0004
0.0023	-	ND < 0.0003	ND < 0.0009
0.0036	-	0.0058	ND < 0.002
0.023	-	0.0038	0.0052
ND < 0.0004	-	0.0055	-
-	-	ND < 0.0004	-

## Data Availability

The data that support the findings of this study are available from the corresponding authors upon reasonable request.

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
