# Peer review of "Linking Pollution and Viral Risk: Detection of Dioxins and Coronaviruses in Cats and Dogs"

_viruses, 2025, doi:10.3390/v17091271_

Round 1

Reviewer 1 Report

Comments and Suggestions for Authors

Comments:
The manuscript by Serra et al. investigated a relevant topic, exploring the potential link between environmental pollution (dioxins and PCBs) and coronavirus infections in cats and dogs. The study combines viral and chemical analyses, providing insight into the possible relationship between environmental contaminants and viral susceptibility. This study addresses a significant area of concern, linking environmental pollution with the health of domestic animals, which could have broader implications for human health. The observation that high levels of 2,3,7,8-TCDD may promote CCoV replication in dogs is intriguing and deserve further investigation. This is a good study. However, I still have concerns to be addressed.

Major concerns:

  1. If authors could confirm the findings of this study with targeted in vitro or in vivo experiments would significantly strengthen the conclusions and provide more compelling evidence for the proposed link between environmental pollution and viral infections.
  2. A table summarizing the observed significant correlations, including the correlation coefficients (r or rs) and p-values, would greatly enhance the clarity of the findings.
  3. The introduction needs to provide a clearer rationale for focusing on dioxins and PCBs. Elaborate on their known mechanisms of immunotoxicity and their relevance to viral infections. Suggest adding a paragraph summarizing the known immunotoxic effects of dioxins and PCBs, and how these effects might increase susceptibility to viral infections.

Minor concerns:

  1. No FCoV-CCoV isolation result was shown in this manuscript.
  2. In Results 3.3, the authors stated that" For PCDD/F and DL-PCB, the congener concentrations were reported in Table 5". However, Table 5 only shows the concentrations of the congener 2,3,7,8-TCDD, not PCDD/F and DL-PCB. This is a mistake.
  3. Similarly, in Results 3.3, the authors stated that "For NDL-PCB, individual congeners concentration, (see Table 6) were expressed in ng/g w.w. and summed". Please check if the results make sense. If the concentration of individual congeners are summed, the unit shall remain as ng/g w.w.

Author Response

Major concerns:

  1. If authors could confirm the findings of this study with targeted in vitro or in vivo experiments would significantly strengthen the conclusions and provide more compelling evidence for the proposed link between environmental pollution and viral infections.

We really appreciate reviewer’s advice. As a matter of fact, we tried to isolate the field viruses detected with the aim of carrying out the experiments suggested. Unfortunately, the isolation failed, and we couldn’t therefore use the strains for in vitro experiments. As to in vivo, we are not authorized to perform this kind of experiment. However, in our previous studies, as better specified at Page 2, Line 91, we have been recently observed that, during infection with canine coronavirus type II in A72 cells, exposure to 2,3,7,8-TCDD enhances CCoV replication in vitro,

  1. A table summarizing the observed significant correlations, including the correlation coefficients (r or rs) and p-values, would greatly enhance the clarity of the findings.

We wish to thank the Referee for this comment. We, however, would prefer to leave the data in the text without adding another table.

  1. The introduction needs to provide a clearer rationale for focusing on dioxins and PCBs.

As suggested, we specified it. Please, see Page 3, Lines 104-119

  1. Elaborate on their known mechanisms of immunotoxicity and their relevance to viral infections. Suggest adding a paragraph summarizing the known immunotoxic effects of dioxins and PCBs, and how these effects might increase susceptibility to viral infections.

We added a paragraph, please, see Page 2, Lines 83-92

Minor concerns:

  1. No FCoV-CCoV isolation result was shown in this manuscript.

A paragraph describing virus isolation results was added in the manuscript (see new par. 3.3)

  1. In Results 3.3, the authors stated that" For PCDD/F and DL-PCB, the congener concentrations were reported in Table 5". However, Table 5 only shows the concentrations of the congener 2,3,7,8-TCDD, not PCDD/F and DL-PCB. This is a mistake.

We are sorry for the mistake. The order of the tables was corrected

  1. Similarly, in Results 3.3, the authors stated that "For NDL-PCB, individual congeners concentration, (see Table 6) were expressed in ng/g w.w. and summed". Please check if the results make sense. If the concentration of individual congeners are summed, the unit shall remain as ng/g w.w.

The unit was revised, and “w.w.” was added to the sum of the individual congener concentrations.

Reviewer 2 Report

Comments and Suggestions for Authors

This is the review of the Manuscript Entitled: “Linking Pollution and Viral Risk: Detection of dioxins and 2 coronaviruses in cats and dogs” by Serra et al. This manuscript presents data demonstrating a correlation between dioxin levels in dogs and cats and infections with Feline Coronavirus (FCoV) and Canine Coronavirus (CCoV). The study reports the presence of coronavirus infections, mildly elevated dioxin levels, and a weak correlation between the two. The manuscript provides a clear description of the methodologies used; however, improvements are needed in the presentation of results and the discussion section. Overall, the study addresses an important and timely topic, with potential implications for environmental health and veterinary science. The authors are encouraged to address the concerns listed below to further strengthen the manuscript.

Major concerns:

Major Concerns:

  1. Section 3.1: Missing data for FCoV viral isolation.
  2. Sensitivity and Accuracy of ddPCR:
    The PCR Ct values for FCoV and CCoV ranged from 22-34 and 23-37, respectively, but the highest viral loads by ddPCR were significantly different: 358 vs. 77.6 copies/µL. Please address this discrepancy and clarify the inconsistency in viral load quantification between PCR and ddPCR methods.
  3. Line 347-348:
    The manuscript mentions Cytopathic Effects (CPE) in cell cultures without proper confirmation. The possible reasoning for this observation is not discussed. Please elaborate on the potential causes of CPE and how they were verified or ruled out in the study.
  4. Discussion Expansion:
    The authors should expand the discussion to explore possible mechanisms underlying the weak correlation between dioxin levels and coronavirus infections. Additionally, addressing the limitations of the study—such as sample size, geographic scope, or potential confounding variables—would strengthen the manuscript.

Minor Concerns:

  1. Line 196-197: Please specify which cell line was used for each virus (FCoV and CCoV).
  2. Line 308: The term "system" should be corrected to “systemic” for clarity.
  3. Throughout the Manuscript: The use of the word “gender” should be replaced with “sex” for greater accuracy in biological and veterinary contexts.
  4. Table 2:The percentage for 11 junior/adult cats is 17.7%, not 18.3%. Please correct this.
  5. Table 3: The table legend indicates FCoV, but the data in the table corresponds to CCoV. Please correct the legend or ensure the data matches the description.
  6. Line 375: The sentence refers to cats, but Table 4 is for dogs. Please check and ensure consistency between the text and table.
  7. Table 4, Line 383 and Lines 476-477: The percentage for 11 junior/adult and young dogs is incorrect. The correct percentages should be 12.2% and 58.3%, respectively. Additionally, please verify the values for the 95% confidence intervals.

Author Response

Major Concerns:

  1. Section 3.1: Missing data for FCoV viral isolation.

Paragraphs describing viruses isolation method (see par. 2.2.6) and results  (see 3.3) were added in the manuscript

  1. Sensitivity and Accuracy of ddPCR: The PCR Ct values for FCoV and CCoV ranged from 22-34 and 23-37, respectively, but the highest viral loads by ddPCR were significantly different: 358 vs. 77.6 copies/µL. Please address this discrepancy and clarify the inconsistency in viral load quantification between PCR and ddPCR methods.

We are grateful to the reviewer for this comment and tried to address (find it below) his question on discrepancy.

In our study we carried out quantification only by ddPCR and not by qPCR. Ct values alone cannot be compared to quantification, since they can be influenced by many factors. We considered Ct values only to discriminate between positive/negative samples and to choose the samples to destinate to viral isolation (Ct<30). Anyway, it is widely reported in the literature that qPCR (made by standard curves using Ct values of a ten-fold diluted standard) has different sensitivity, precision with respect to ddPCR (see literature below reported)

  • Te SH, Chen EY, Gin KY. 2015. Comparison of Quantitative PCR and Droplet Digital PCR Multiplex Assays for Two Genera of Bloom-Forming Cyanobacteria, Cylindrospermopsis and Microcystis . Appl Environ Microbiol 81
  • Morón-López S, Riveira-Muñoz E, Urrea V, Gutiérrez-Chamorro L, Ávila-Nieto C, Noguera-Julian M, Carrillo J, Mitjà O, Mateu L, Massanella M, Ballana E, Martinez-Picado J. 2024. Correction for Morón-López et al., “Comparison of Reverse Transcription (RT)-Quantitative PCR and RT-Droplet Digital PCR for Detection of Genomic and Subgenomic SARS-CoV-2 RNA”. Microbiol Spectr 12:e02269-24.
  • Park C, Lee J, Hassan ZU, Ku KB, Kim SJ, Kim HG, Park EC, Park GS, Park D, Baek SH, Park D, Lee J, Jeon S, Kim S, Lee CS, Yoo HM, Kim S. Comparison of Digital PCR and Quantitative PCR with Various SARS-CoV-2 Primer-Probe Sets. J Microbiol Biotechnol. 2021 Mar 28;31(3):358-367. doi: 10.4014/jmb.2009.09006. PMID: 33397829; PMCID: PMC9705847.

  1. Line 347-348: The manuscript mentions Cytopathic Effects (CPE) in cell cultures without proper confirmation. The possible reasoning for this observation is not discussed. Please elaborate on the potential causes of CPE and how they were verified or ruled out in the study.

A brief discussion was added in discussion section as requested (see lines 516-518)

  1. Discussion Expansion: The authors should expand the discussion to explore possible mechanisms underlying the weak correlation between dioxin levels and coronavirus infections. Additionally, addressing the limitations of the study—such as sample size, geographic scope, or potential confounding variables—would strengthen the manuscript.

We wish to thank the Referee for this comment. As suggested, we specified them Please, see Pages 16-17, Lines 561-575.

Minor Concerns:

  1. Line 196-197: Please specify which cell line was used for each virus (FCoV and CCoV).

Cell lines were specified for each virus

  1. Line 308: The term "system" should be corrected to “systemic” for clarity.

The term “system” was corrected to “systemic”

  1. Throughout the Manuscript: The use of the word “gender” should be replaced with “sex” for greater accuracy in biological and veterinary contexts.

The word “gender” was replaced by “sex” as suggested

  1. Table 2: The percentage for 11 junior/adult cats is 17.7%, not 18.3%. Please correct this.

The correction was made as suggested

  1. Table 3: The table legend indicates FCoV, but the data in the table corresponds to CCoV. Please correct the legend or ensure the data matches the description.

The legend was corrected according to reviewer’s suggestion

  1. Line 375: The sentence refers to cats, but Table 4 is for dogs. Please check and ensure consistency between the text and table.

The sentence was corrected

  1. Table 4, Line 383 and Lines 476-477: The percentage for 11 junior/adult and young dogs is incorrect. The correct percentages should be 12.2% and 58.3%, respectively. Additionally, please verify the values for the 95% confidence intervals.

The number of junior/adult was corrected to 39. Percentages were modified as indicated.

As correctly suggested 95% confidence intervals were verified.

Reviewer 3 Report

Comments and Suggestions for Authors

The full report is attached.

Comments on the Quality of English Language

The manuscript would benefit from careful editing by a fluent English speaker or a professional language editing service. While the general meaning is understandable, there are several grammatical errors, awkward phrasings, and inconsistent use of scientific terminology that detract from the clarity and impact of the work. Specific issues include:

-Unnatural or overly complex sentence constructions that hinder readability.

-Inconsistent use of verb tenses (particularly in the Methods and Results sections).

-Occasional misuse of technical terms (e.g., “polluted animals,” “positive group”), which may be confusing or imprecise for readers.

-Overuse of vague language (e.g., “high values,” “affected animals”) instead of clear, specific descriptors.

Improving the English language quality would significantly enhance the professionalism and accessibility of the manuscript.

Author Response

  1. The manuscript would benefit from careful editing by a fluent English speaker or a professional language editing service. While the general meaning is understandable, there are several grammatical errors, awkward phrasings, and inconsistent use of scientific terminology that detract from the clarity and impact of the work. Specific issues include: Unnatural or overly complex sentence constructions that hinder readability. -Inconsistent use of verb tenses (particularly in the Methods and Results sections).

As correctly suggested by the reviewer the whole paper was edited by a fluent English speaker.

  1. Occasional misuse of technical terms (e.g., “polluted animals,” “positive group”), which may be confusing or imprecise for readers.

Inappropriate use of technical terms was revised throughout the paper. As an example, the terms positive and negative were substituted with more appropriate infected and non-infected

  1. Overuse of vague language (e.g., “high values,” “affected animals”) instead of clear, specific descriptors. Improving the English language quality would significantly enhance the professionalism and accessibility

As correctly suggested by the reviewer the whole paper was edited by a fluent English speaker.

  1. The criteria for classifying animals into the different groups (especially "affected by coronavirus" vs. “not affected”) are unclear and inconsistently described.

Explanation of the term “affected” was clarified in the Tables 1 and 3

  1. What viral targets were used in PCR? Was typing/sequencing performed to confirm whether detected CoVs were enteric, respiratory, or potentially zoonotic?

We added, as requested, in paragraph 2.2.2 and 2.2.3 viral targets used in real-time PCR. As indicated in the paper only typing of CCoV was performed (see par.2.2.4). Sequencing (by Sanger) of field samples was unsuccessful (data not shown) probably because of complex matrixes. As explained throughout the manuscript the coronavirus detected are enteric, but capable of causing respiratory/systemic diseases. Their zoonotic potential is unknown but, as reported in the manuscript (see lines 103-104), there are cases in literature describing variants infecting humans

  1. How were dioxin-exposed animals selected, and what defines "polluted area"? No geographic coordinates or exposure thresholds are discussed.

As better explained in the paper (see par. 2.1) the animals were not specifically selected for the study by their carcasses were sent to our institute following a regional plan. As better explained in the manuscript (see lines 108-109) the polluted areas have been classified in a previous study (see Pizzolante et al. 2021)

  1. Define inclusion/exclusion criteria more explicitly. Include animal lifestyle information (indoor/outdoor), diet (commercial/homemade), vaccination status, and age distribution. These are crucial confounders.

Information about lifestyle was accessible only to distinguish stray from owned animals. No more data (indoor/outdoor) was available. Diet and vaccination status were not indicated by Veterinarians collecting the animals. Age classification (Junior/adult and kitten or young) was reported in Tables 2 and 4.

  1. The sample size is too small to support any statistically meaningful conclusions, especially with subgroup breakdowns (e.g., only 3 cats in one group).   No confidence intervals, p-values, or correlation analyses are reported between pollutant levels and viral status. Include appropriate non-parametric or categorical tests (e.g., Fisher’s exact, Mann–Whitney U) and clarify how sample sizes were justified.

As described in Section 2.4 (Statistical Analysis), non-parametric tests were performed to investigate the relationships among the variables (congener concentrations and CoVs positivity). Specifically, the Mann–Whitney U test was used to determine whether the concentrations of PCDD/F (including 2,3,7,8-TCDD), DL-PCB, and NDL-PCB differed significantly between CoV-positive and CoV-negative cats and dogs. In addition, the non-parametric Spearman test was employed to evaluate correlations between each congener of PCDD/F, DL-PCB, NDL-PCB, their sums, and CoV positivity. A p-value of ≤ 0.05 was considered statistically significant in all tests.

The results of the statistical analyses, including p-values and correlation coefficients, are reported in Section 3.4 (lines 473–479) and commented in the discussion (lines 546-548): “Our results showed in dogs a moderate but significant correlation between chemical exposure and increased susceptibility to CCoV infection (Spearman’s rs=0.59; p=0.031)”.

  1. The conclusions claim that animals with higher pollutant loads are more likely to be CoVpositive, but this is not statistically or mechanistically demonstrated. Statements such as “pollution could be a risk factor for viral infection” are not justified based on correlative presence/absence data alone. Temper language throughout the abstract and discussion to reflect the observational, hypothesis-generating nature of the findings.

Language was tempered as requested by substituting,

in the abstract: “May promote” with “may be associated”

In the discussion and conclusion section respectively: “may influence” with “maybe somehow associated”, “confirming” with “hypothesizing”

  1. There are no Ct values, gel images, or information on PCR controls provided.

Detection sensitivity and specificity are unknown. The type of coronavirus (e.g., FCoV, CCoV, SARS-related) is not described, which is essential given the study’s implications for public health. Describe PCR targets, controls, and validation.

As correctly suggested by the reviewer, more information was added on PCR targets, controls and validation of the assay in the new paragraphs 2.2.2 and 2.2.3.

  1. If no typing was performed, state this and discuss the limitations.

As already described in the material and methods (see par. 2.2.4) as well as in the results (see par. 3.2) typing was carried out only to distinguish CCoV in I and II. Real time positive samples (both to CCoV and FCoV) underwent viral isolation, which was unsuccessful; therefore, further genetic characterization on detected strains was not possible.

  1. Table 1 is difficult to read and lacks units. The grouping and labeling of animals are not intuitive. The figure summarizing dioxin levels is helpful but lacks detailed legend explanation and statistical annotation. Reformat tables for clarity. Include individual data points (scatter/dot plots) instead of bar graphs with standard error, especially given the low n.

Table 1 and 3 were reformatted for better clarity as requested. As to the figure, it represents viral distribution in the organs. This was better clarified in the figure legend.

  1. Use consistent terminology: Avoid switching between “polluted group,” “positive group,” “sick animals,” etc. Several grammatical errors need correction (e.g., “Among the groups, differences can be evidenced...”). Clarify whether the observed clinical signs were selfreported by owners or veterinarian-confirmed.

In Tables 1 and 3 we report anatomopathological findings described by our veterinarians during necropsy. No information about clinical signs in life was available since animals were found already dead by veterinary official service and sent to our institute for necropsy and further investigations on death cause. This was better clarified in the materials and methods section (see par. 2.1)

  1. The introduction should briefly explain known immunosuppressive effects of dioxins in mammals (e.g., via AhR pathway), to provide a stronger mechanistic basis.

The explanation was added in the discussion section (see lines 542-550)

Reviewer 4 Report

Comments and Suggestions for Authors

I have finished the review of the manuscript entitled "Linking Pollution and Viral Risk: Detection of dioxins and coronaviruses in cats and dogs" the approach is novel and opens discussion to a relevant topic.

Nevertheless some aspects need further clarification.

Percentage match to previously published papers is 45% according to iThenticate, even when previous work by authors is correctly cited, rephrasing coul improve originality, please consider editing.

Keywords are accurate, but there is still place for adding "veterinary" or similar keyword to correctly differentiate from human cases.

The manuscript is largely compliant with ARRIVE 2.0, especially in methodology, results, and transparency. ARRIVE guidelines are thye most appropiate for animal studies as in this case. 

Sample is presented, including number of animals (80 cats, 51 dogs) clearly stated, but there is no justification or explanation on how sample size was calculated/ estimated.

Exclusion criteria not explicitly stated, though some samples excluded due to inconclusive viral culture, should this be considered as exclusion? or elimination?

It should be mentioned if blinding in necropsy, PCR, or chemical analyses. 

Some sentences are overly long, making sections dense (e.g., background on PCDD/PCDFs spans several clauses without break). Breaking them into shorter sentences would improve readability and flow of themes.

Grammar flaws such as: “met with contrasting opinions” could be “has met contrasting results in the literature”

Maybe shortening long sentences would be useful as much as using active voice more often, and vary connectors.çç

I would also reccomend to avoid redundancy in Results and Discussion; streamline pollutant background in Introduction.

Some interpretative language leans toward causal inference (“high levels of TCDD may promote CCoV infection”) even though evidence is correlational. Please use cautious tone (“associated with” instead of “promote”) when interpreting correlations.

Add a specific “Limitations” subsection for aknowleding potential biases (sampling bias, lack of randomization/blinding, observational nature)

Author Response

  1. Percentage match to previously published papers is 45% according to iThenticate, even when previous work by authors is correctly cited, rephrasing coul improve originality, please consider editing.

The paper was accurately rephrased where possible trying to improve originality.

  1. Keywords are accurate, but there is still place for adding "veterinary" or similar keyword to correctly differentiate from human cases.

Keyword “veterinary viruses” was added

  1. Sample is presented, including number of animals (80 cats, 51 dogs) clearly stated, but there is no justification or explanation on how sample size was calculated/ estimated.

More information was added in the paragraph 2.1

  1. Exclusion criteria not explicitly stated, though some samples excluded due to inconclusive viral culture, should this be considered as exclusion? or elimination?

We better clarified in the paper (see par. 3.3) that positivity to coronavirus was ascertained by Real Time RT-PCR assays. Viral culture was carried out only with the aim to further study (in vitro experiments) and characterize (sequencing), the field viral strains. Consequently, inconclusive viral culture didn’t lead to any sample exclusion

  1. It should be mentioned if blinding in necropsy, PCR, or chemical analyses. 

We better explained in the manuscript the criteria by which samples were chosen. Necropsy and PCR were blind, only chemical analysis was decided on the basis of PCR results.

  1. Some sentences are overly long, making sections dense (e.g., background on PCDD/PCDFs spans several clauses without break). Breaking them into shorter sentences would improve readability and flow of themes. Grammar flaws such as: “met with contrasting opinions” could be “has met contrasting results in the literature” Maybe shortening long sentences would be useful as much as using active voice more often, and vary connectors.çç I would also reccomend to avoid redundancy in Results and Discussion; streamline pollutant background in Introduction.

The paper was fully revised, and sentences were shortened (where possible) as suggested

  1. Some interpretative language leans toward causal inference (“high levels of TCDD may promote CCoV infection”) even though evidence is correlational. Please use cautious tone (“associated with” instead of “promote”) when interpreting correlations.

Following reviewer’s advice, the language was mitigated throughout the manuscript.

  1. Add a specific “Limitations” subsection for aknowleding potential biases (sampling bias, lack of randomization/blinding, observational nature)

“Limitations” were added in the conclusion section as requested.

Round 2

Reviewer 1 Report

Comments and Suggestions for Authors

No further comment.

Author Response

Figures and tables were improved.

Reviewer 3 Report

Comments and Suggestions for Authors

The report is attached.

Comments on the Quality of English Language

The revised manuscript demonstrates substantial improvement in English language quality. The authors have corrected many grammatical issues, improved sentence structure, and clarified ambiguous terminology. A few minor stylistic edits could still enhance readability, particularly in the Discussion section, but the current version is acceptable for publication without additional editing.

Author Response

  1. The use of naturally deceased animals within a regional monitoring program is appropriate and ethically justified. However, a short schematic figure or flowchart describing group allocation and sampling would further improve reader clarity. The authors could consider emphasizing early in the Introduction that this is a post-mortem tissue-based study, not a longitudinal or clinical surveillance effort.

As correctly indicated by the reviewer, we emphasized in the Introduction section that the research was based on a post-mortem tissue study (lines 112-113).

  1. The inclusion of specific PCR targets, validation steps, and clarification that only CCoV typing was performed (with sequencing attempts unsuccessful) improves the transparency of viral detection methods. Still, it may be valuable to state explicitly in the Discussion that inability to type most viruses represents a key limitation.

Limitation due to unsuccessful viral sequencing was specified (see lines 605-606) as suggested.

  1. Tables 1 and 3 are clearer, and the revised terminology is more consistent. Consider using

bold headers or color shading to further improve visual separation between

infected/noninfected groups. Clarify in Table legends whether necropsy signs were collected

solely post-mortem or if any clinical histories were available.

All the tables were improved following suggestion. Furthermore, we clarified in the legends  of tables 1 and 3 what suggested.

Reviewer 4 Report

Comments and Suggestions for Authors

The revised review of the manuscript "Linking Pollution and Viral Risk: Detection of dioxins and coronaviruses in cats and dogs" the actual version is improved and covers clearly aspects that were not clearly explained before.

Percentage match reduced.

It is now presented as a veterinary research

Sample size is clear now, language is more objective and limitations are presented.

I recommend the present version for publication.

Author Response

The manuscript was improved.